# Entropic singularities give rise to quantum transmission

Vikesh Siddhu [1,2] ✉

When can noiseless quantum information be sent across noisy quantum devices? And at what maximum rate? These questions lie at the heart of quantum technology, but remain unanswered because of non-additivity— a fundamental synergy which allows quantum devices (aka quantum channels) to send more information than expected. Previously, non-additivity was known to occur in very noisy channels with coherent information much smaller than that of a perfect channel; but, our work shows non-additivity in a simple low-noise channel. Our results extend even further. We prove a general theorem concerning positivity of a channel's coherent information. A corollary of this theorem gives a simple dimensional test for a channel's capacity. Applying this corollary solves an open problem by characterizing all qubit channels whose complement has non-zero capacity. Another application shows a wide class of zero quantum capacity qubit channels can assist an incomplete erasure channel in sending quantum information. These results arise from introducing and linking logarithmic singularities in the von-Neumann entropy with quantum transmission: changes in entropy caused by this singularity are a mechanism responsible for both positivity and non-additivity of the coherent information. Analysis of such singularities may be useful in other physics problems.

[1] Department of Physics, Carnegie Mellon University, Pittsburgh, Pennsylvania 15213, USA. [2] Present address: JILA, University of Colorado/NIST, Boulder, CO 80309, USA. ✉email: vsiddhu@protonmail.com

Entropy is fundamental. As a measure of complexity in a statistical distribution, entropy is widely used in learning theory[1,2], economics[3,4], and cryptography[5]. In physics, entropy usually quantifies disorder. It is used to express laws of thermodynamics[6–8], explore the nature of black holes[9–11], and study a variety of other physical phenomenon[12–18]. Advances in understanding mathematical and computational properties of entropy[19–22] have opened the doors for deeper insights in physics and many other areas of study.

One area where entropy provides key insights is information science[23]. The Shannon entropy not only quantifies the amount of classical information in a source, it also plays a fundamental role in answering a key practical question: when, and at what maximum rate can classical information be sent across noisy communication channels? The maximum Shannon entropy common between a channel's input and output, called the channel mutual information $C^{(1)}$, gives an achievable rate at which error correcting codes can recover noiseless information sent across many uses of a noisy channel.

The channel mutual information satisfies a crucial property, additivity: the channel mutual information for two channels used together is the sum of each. This additivity ensures that the channel capacity $C$, defined as the best possible achievable rate, simply equals the channel mutual information $C^{(1)}$. More remarkably, additivity implies that the channel capacity completely specifies a classical channel's ultimate ability to send information. These implications are not only fundamental to our understanding of noisy classical information but also critical to the use of channel capacity as a benchmark for error correcting codes. These codes are essential for storing and sending noiseless classical information across noisy channels[24,25].

The physical world is not classical but quantum mechanical. It contains quantum information, which is strikingly different from its classical counterpart[26–29]. In practice, noisy quantum devices carry quantum information. These devices, which may send, store, or process information, are modeled mathematically by completely positive trace-preserving maps, also called (noisy) quantum channels. While quantum information can be extremely useful for computing and communication, it is notoriously error prone. Consequently, there are both fundamental and practical reasons to understand when and at what maximum rate can noiseless quantum information be stored, processed, or sent across noisy quantum channels[30–36]. Despite dedicated efforts, there is no satisfactory answer to this basic question. The key reason behind this unsatisfactory state of affairs is nonadditivity in the quantum analog[37,38], $Q^{(1)}$, of the channel mutual information $C^{(1)}$: for two noisy quantum channels $\mathcal{B}_1$ and $\mathcal{B}_2$ used in parallel, the channel coherent information $Q^{(1)}$ satisfies an inequality,

$$Q^{(1)}(\mathcal{B}_1 \otimes \mathcal{B}_2) \geq Q^{(1)}(\mathcal{B}_1) + Q^{(1)}(\mathcal{B}_2), \tag{1}$$

which can be strict[39]. Like $C^{(1)}$, $Q^{(1)}$ is an entropic quantity, however, it represents an achievable rate for correcting errors in quantum information sent across a noisy quantum channel. A channel's quantum capacity $Q$ is defined to be the best possible achievable rate[30]. Nonadditivity of $Q^{(1)}$ makes $Q$ difficult to compute[40,41], and more markedly it makes $Q$ an incomplete measure[42] of a channel's ability to send quantum information.

The difficulty in computing $Q$ essentially comes from a strict inequality in (1), found when $\mathcal{B}_1$ and $\mathcal{B}_2$ are tensor products of the same channel $\mathcal{B}$. Low–dimensional channels which display this type of nonadditivity include a variety of very noisy qubit channels including the depolarizing[39,43], the dephrasure[44] and other qubit Pauli[43,45,46] and generalized erasure channels[47,48]. As a result, even when a channel $\mathcal{B}$ is relatively simple, its quantum

capacity $Q(\mathcal{B})$ must be obtained as the limit $n \mapsto \infty$ of a sequence $Q^{(1)}(\mathcal{B}^{\otimes n})/n$[32–36]. This limit, sometimes called a regularization of $Q^{(1)}$, can be particularly intractable: there are very noisy high dimensional channels for which each term in this sequence can be larger than the previous one[49]. In addition, for any integer $k$ there is a channel $\tilde{\mathcal{B}}$ for which $Q^{(1)}(\tilde{\mathcal{B}}^{\otimes k}) = 0$ but $Q(\tilde{\mathcal{B}}) > 0$[50]. This type of unbounded nonadditivity makes it hard to even check if a channel's quantum capacity is strictly positive or zero.

Challenges in computing and checking the positivity of a channel's quantum capacity can be circumvented in the special case of (anti-)degradable channels[51,52], PPT channels[53], DSPT channels[54], and less noisy channels[55]. However, even if one computes a channel's quantum capacity, nonadditivity implies that this capacity may be an incomplete measure of the channel's ability to send quantum information. Instances of nonadditivity, i.e., a strict inequality in (1), have been found when $\mathcal{B}_1$ and $\mathcal{B}_2$ are different channels, each having no quantum capacity. One instance, called *superactivation* has been found when $\mathcal{B}_1$ is a PPT channel and $\mathcal{B}_2$ is a zero capacity erasure or depolarizing channel[42,56]. Another instance of nonadditivity has been found where $\mathcal{B}_1$ is a rocket channel and $\mathcal{B}_2$ is an erasure channel[57], both channels are again very noisy, $\mathcal{B}_1$ has small quantum capacity while $\mathcal{B}_2$ has none, but together they have coherent information much larger than the sum of quantum capacities of each channel.

In the past, instances of nonadditivity found in very noisy channels have shown that quantum information and channels can display a type of synergy which is absent from their classical counterparts. Nonadditivity has previously not been found in low-noise channels, those with coherent information comparable to the quantum capacity of a perfect (identity) channel with the same input dimension as the channel. By contrast, in certain low-noise channels nonadditivity has been shown to be absent[58], and in low-noise Pauli channels nonadditivity has been shown to be of little practical relevance[59]. While the study of nonadditivity remains of fundamental interest, methods for finding and exploring nonadditivity are scant. In high dimensional and high noise PPT and rocket channels, nonadditivity is found by using the special structure of these channels. Whereas in qubit and other low dimensional but high noise channels, methods based on degenerate quantum codes[45] and numerical searches[60] can identify nonadditivity, but these too can falter in simple cases of interest[47].

Strategies to check if a general channel has zero or nonzero quantum capacity are limited[61,62]. To test if a channel has zero quantum capacity, one can check whether the channel is PPT or anti-degradable. For special channels these two checks can be done algebraically[52,62,63], but in general, they require numerically solving a semi-definite program[64]. Even if one performs these checks, their results can be inconclusive because there may exist zero capacity channels that are neither PPT nor anti-degradable. Testing if a channel has nonzero quantum capacity is tricky. Except in very special circumstances, there are no algebraic tests. Numerics can be used to check if a channel's coherent information is nonzero. However, these numerics can be unreliable, even for low-dimensional qubit channels[44,65,66] without unbounded nonadditivity. For high dimensional channels numerics can be expensive[43,67] in addition to being unreliable[50].

Seeking physical and mathematical mechanisms to find and understand positivity and nonadditivity remains an enduring challenge in quantum information science. While this challenge tempers hopes for rapid progress on understanding quantum capacities, it also presents an opportunity to introduce new ideas for addressing this challenge.

In this work, we introduce a simple but key property of the von-Neumann entropy, which we call a log-singularity

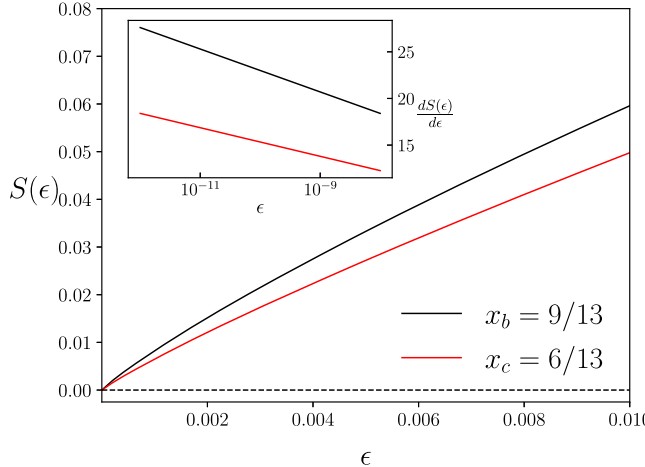

**Fig. 1 Behaviour of the von-Neumann entropy in the vicinity of an $\epsilon$ log-singularity.** Two density operators $\rho_b(\epsilon)$ and $\rho_c(\epsilon)$ with spectrum $(1 - x_b\epsilon, x_b\epsilon)$ and $(1 - x_c\epsilon, x_c\epsilon/3, x_c\epsilon/3, x_c\epsilon/3)$ respectively have entropies $S_b(\epsilon)$ and $S_c(\epsilon)$ respectively, where $0 \leq \epsilon \leq 1$. For fixed $\epsilon$ log-singularity rates $x_b = 9/13$ and $x_c = 6/13$, of $S_b(\epsilon)$ and $S_c(\epsilon)$, respectively, a plot of these entropies $S(\epsilon)$ as a function of $\epsilon$. The inset shows the gradient of the entropies as a function of $\log_2 \epsilon$ for small $\epsilon$.

(see Fig. 1), and show that changes in entropy are caused by log-singularities are a mathematical mechanism responsible for both positivity and nonadditivity of the coherent information. Utilizing this mechanism, (1) we provide an instance of nonadditivity using a zero capacity qubit channel in parallel with a low-noise qutrit channel with $\mathcal{Q}^{(1)}/\log_2 3 \simeq 0.6$; (2) we prove a general theorem which gives algebraic conditions under which a quantum channel must have strictly positive coherent information. A corollary of this theorem gives a simple, dimensional test for capacity. An application of this corollary provides a characterization of all qubit channels whose complement has nonzero quantum capacity. A separate application of the theorem reveals how a large class of zero capacity qubit channels can assist an incomplete erasure channel in sending quantum information.

## Results

**Log-singularity**. Let $\rho(\epsilon)$ denote a density operator that depends on a real positive parameter $\epsilon$, and $S(\epsilon) = -\mathrm{Tr}(\rho(\epsilon)\log\rho(\epsilon))$ denote its von-Neumann entropy. If one or several eigenvalues of $\rho(\epsilon)$ increase linearly from zero to leading order in $\epsilon$ then a small increase in $\epsilon$ from zero increase $S(\epsilon)$ by $x|\epsilon\log\epsilon|$ for some constant $x > 0$; i.e., $dS(\epsilon)/d\epsilon \simeq -x\log\epsilon$, and we say $S(\epsilon)$ has an $\epsilon$ log-singularity with rate $x$. For instance, a qubit density operator with spectrum $(1 - x_b\epsilon, x_b\epsilon), 0 \leq \epsilon \leq 1$ and $0 \leq x_b \leq 1$ has an $\epsilon$ log-singularity of rate $x_b$. A quart density operator with spectrum $(1 - x_c\epsilon, x_c\epsilon/3, x_c\epsilon/3, x_c\epsilon/3)$ and $0 \leq x_c \leq 1$, has an $\epsilon$ log-singularity of rate $x_c$ (also see Fig. 1).

The term $\epsilon$ log-singularity comes from the behavior where the derivative of $S(\epsilon)$ with respect to $\epsilon$ is logarithmic in $\epsilon$ and this derivative tends to infinity as $\epsilon$ tends to zero. While $S(\epsilon)$ is continuous, the behavior of continuity bounds on $S(\epsilon)$ can be dominated by $\epsilon$ log-singularities in the sense that changes in continuity bounds can be essentially logarithmically in $\epsilon$ for small $\epsilon$ (see Supplementary Note 2). For very small $\epsilon$, this singularity causes a sharp change in the von-Neumann entropy. Since this sharp change occurs for very small parameter values $\epsilon$, its effects can be prohibitively hard to detect numerically. However, when these effects appear, they dominate the behavior of the von-Neumann entropy and the physics which may directly depend on this entropy.

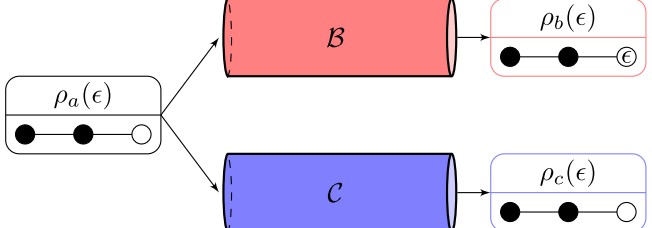

**Fig. 2 Schematic for the reason behind an $\epsilon$ log-singularity.** An input density operator $\rho_a(\epsilon)$ is mapped by a channel $\mathcal{B}$ to a density operator $\rho_b(\epsilon)$ and by the channel's complement $\mathcal{C}$ to a different density operator $\rho_c(\epsilon)$. Below each operator is a representation of its spectrum where closed and open circles indicate eigenvalues that, for all $0 \leq \epsilon \leq 1$, are nonzero and zero, respectively. A circle with $\epsilon$ indicates an eigenvalue that increases linearly from zero to leading order in $\epsilon$. Since the spectrum of $\rho_b(\epsilon)$ has a circle with $\epsilon$, its von-Neumann entropy has an $\epsilon$ log-singularity.

Understanding of the physics of sending noiseless quantum information across a noisy quantum channel is aided by the channel's coherent information $\mathcal{Q}^{(1)}$. To define a quantum channel and its coherent information, consider an isometry $J: a \mapsto b \otimes c$ that generates a pair of quantum channels $\mathcal{B}: a \mapsto b$ and $\mathcal{C}: a \mapsto c$, where each channel may be called the complement of the other. These channels map an input density operator $\rho_a$ to outputs $\rho_b := \mathcal{B}(\rho_a) = \mathrm{Tr}_c(J\rho_a J^\dagger)$ and $\rho_c := \mathcal{C}(\rho_a) = \mathrm{Tr}_b(J\rho_a J^\dagger)$, respectively. The dimensions $d_b$ and $d_c$, of outputs $b$ and $c$, respectively are the ranks of $\mathcal{B}(I_a)$ and $\mathcal{C}(I_a)$, respectively. These are the smallest possible output dimensions required to define $\mathcal{B}$ and $\mathcal{C}$ (in the notation of Def. 4.4.4 in[68], $d_b$ is the Choi-rank of $\mathcal{C}$ and $d_c$ is the Choi-rank of $\mathcal{B}$). These definitions make the channel pair setting symmetric with respect to the replacement of one channel in the pair with its complement. The coherent information (or the entropy bias) of $\mathcal{B}$ at $\rho_a$, $\Delta(\mathcal{B}, \rho_a) := S(\rho_b) - S(\rho_c)$, maximized over density operators $\rho_a$ gives the channel coherent information $\mathcal{Q}^{(1)}(\mathcal{B})$.

When considering an input density operator, $\rho_a(\epsilon)$, we use a concise notation $S_b(\epsilon) := S(\rho_b(\epsilon))$, $S_c(\epsilon) := S(\rho_c(\epsilon))$, and $\Delta(\epsilon) := \Delta(\mathcal{B}, \rho_a(\epsilon)) = S_b(\epsilon) - S_c(\epsilon)$. At $\epsilon = 0$ if $\rho_a(\epsilon)$ has rank $d_a$, then by definition of $d_b$, rank of $\rho_b(0)$ will be $d_b$ (see Supplementary Note 3), as a result, $S_b(\epsilon)$ will not have an $\epsilon$ log-singularity. A similar argument shows if $\rho_a(0)$ is rank $d_a$ then $S_c(\epsilon)$ does not have an $\epsilon$ log-singularity. When the rank of $\rho_a(0)$ is strictly less than $d_a$, then an $\epsilon$ log-singularity can be present in $S_b(\epsilon)$ or $S_c(\epsilon)$ (see Fig. 2), or an $\epsilon$ log-singularity can be present in both $S_b(\epsilon)$ and $S_c(\epsilon)$ in which case the $\epsilon$ log-singularity with larger rate is said to be stronger.

**Positivity of coherent information**. Changes in the von-Neumann entropy caused by log-singularities can act as a mechanism which makes $\mathcal{Q}^{(1)}(\mathcal{B}) > 0$ (see Fig. 3). To illustrate this mechanism consider a convex combination of input density operators $\hat{\rho}_a$ and $\sigma_a$:

$$\rho_a(\epsilon) = (1 - \epsilon)\hat{\rho}_a + \epsilon\sigma_a, \quad \epsilon \in [0, 1]. \quad (2)$$

This convex combination (2) leads to other such combinations,

$$\rho_b(\epsilon) = (1 - \epsilon)\hat{\rho}_b + \epsilon\sigma_b \quad \text{and} \quad \rho_c(\epsilon) = (1 - \epsilon)\hat{\rho}_c + \epsilon\sigma_c, \quad (3)$$

at the outputs of $\mathcal{B}$ and $\mathcal{C}$, respectively. Let $\hat{\rho}_a$ be a pure state, then $\Delta(\mathcal{B}, \hat{\rho}_a) = 0$ i.e., $\Delta(0) = 0$ (see Supplementary Note 3). Assume $\hat{\rho}_a, \sigma_a$, and the channel pair $(\mathcal{B}, \mathcal{C})$ are such that an $\epsilon$ log-singularity is present in $S_b(\epsilon)$ but not in $S_c(\epsilon)$; that is, for a small enough increase in $\epsilon$ from zero, $S_b(\epsilon)$ increases by $|O(\epsilon\log\epsilon)|$ but $S_c(\epsilon)$ has no $O(\epsilon\log\epsilon)$ increase. Thus, for small enough $\epsilon$,

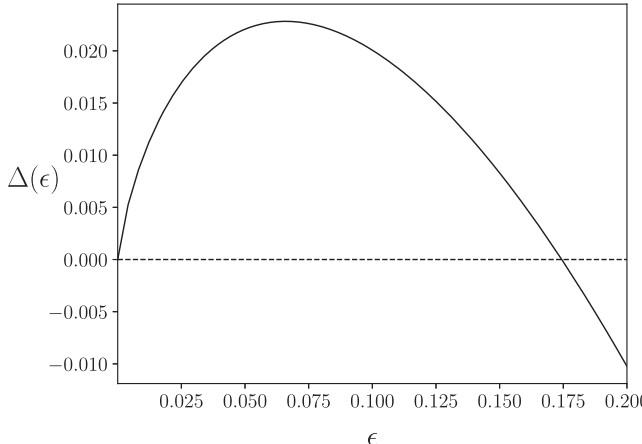

**Fig. 3 Illustration of log-singularity-based mechanism behind positivity of the channel coherent information.** For the channel $\mathcal{B}$ defined by isometry in eq. 4, the entropy difference $\Delta(\epsilon) = S_b(\epsilon) - S_c(\epsilon)$ for density operators below eq. 4 as a function of $\epsilon$ is plotted above. Here $S_b(\epsilon)$ and $S_c(\epsilon)$ have the same entropy at $\epsilon = 0$ and they both have $\epsilon$ log-singularities. The singularity in $S_b(\epsilon)$ has a rate $x_b = 9/13$ which is higher than $x_c = 6/13$, the rate of the singularity in $S_c(\epsilon)$. This higher rate $x_b$ makes both $\Delta(\epsilon)$ and $\mathcal{Q}^{(1)}(\mathcal{B})$ strictly positive for small $\epsilon$, even though for larger $\epsilon$, $\Delta(\epsilon) < 0$.

$$\Delta(\epsilon) \simeq |O(\epsilon \log \epsilon)| > 0; \quad \text{since} \quad \Delta(\epsilon) \le \mathcal{Q}^{(1)}(\mathcal{B}), \quad \text{we conclude } \mathcal{Q}^{(1)}(\mathcal{B}) > 0.$$

In the illustration above, let the channels $\mathcal{B}: a \mapsto b$ and $\mathcal{C}: a \mapsto c$ be defined by an isometry $L: a \mapsto b \otimes c$ of the form,

$$L|0\rangle = |00\rangle, \quad L|1\rangle = \sqrt{\frac{2}{9}}|01\rangle + \sqrt{\frac{7}{9}}|10\rangle, \quad \text{and}$$
$$L|2\rangle = \frac{1}{\sqrt{2}}(|02\rangle + |13\rangle), \tag{4}$$

where $\{|i\rangle\}$ represents the standard basis, and $|ij\rangle$ denotes $|i\rangle \otimes |j\rangle \in b \otimes c$. Let $[\psi]$ denote the dyad $|\psi\rangle\langle\psi|$. A channel input of the form in (2) with $\hat{\rho}_a = [0]$ and $\sigma = (9[1] + 4[2])/13$ leads to channel outputs of the form (3). These outputs $\rho_b(\epsilon)$ and $\rho_c(\epsilon)$ have $\epsilon$ log-singularities with rates $x_b = 9/13$ and $x_c = 6/13$, respectively (see Fig. 1). Since $x_b > x_c$, $\mathcal{Q}^{(1)}(\mathcal{B}) > 0$, as shown in Fig. 3.

In general, a log-singularity-based mechanism can make $\mathcal{Q}^{(1)}(\mathcal{B}) > 0$ if there is a channel input $\rho_a(\epsilon)$ for which $\Delta(0) = 0$, and either there is an $\epsilon$ log-singularity in $S_b(\epsilon)$ but not in $S_c(\epsilon)$ or there is an $\epsilon$ log-singularity in both $S_b(\epsilon)$ and $S_c(\epsilon)$ but the one in $S_b(\epsilon)$ is stronger, in either case, an argument similar to the one in our illustration above implies that $\mathcal{Q}^{(1)}(\mathcal{B}) > 0$. An analogous log-singularity-based mechanism can make $\mathcal{Q}^{(1)}(\mathcal{C}) > 0$. In principle, this mechanism can be applied to a quantum channel $\mathcal{B}$, regardless of how small or large $\mathcal{Q}^{(1)}(\mathcal{B})$ may be. In practice, we find that the above mechanism applies in a general situation presented next.

**Theorem 1**. If a quantum channel $\mathcal{B}$, with output and environment dimension $d_b$ and $d_c$ respectively, maps some pure state to an output of rank $d_c < d_b$, then $\mathcal{Q}^{(1)}(\mathcal{B}) > 0$.

This theorem applies quite generally, including cases where $d_b \ge d_c$. In these $d_b \ge d_c$ cases, a channel $\mathcal{B}$'s coherent information is strictly positive if the theorem holds for any sub-channel of $\mathcal{B}$. There are simple examples (for instance see Supplementary Note 4) of channels with $d_b = d_c$ where the above theorem applies.

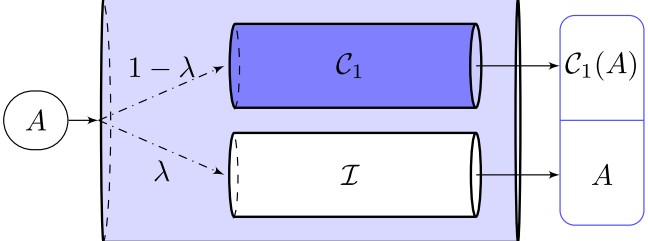

**Fig. 4 Incomplete erasure channel $\mathcal{C}$.** Channel $\mathcal{C}$'s input $A$, with probability $\lambda$ goes via $\mathcal{I}$ (the identity channel) to an output subspace as $A$, or else via $\mathcal{C}_1$ (a noisy channel) to an orthogonal subspace as $\mathcal{C}_1(A)$. When $\mathcal{C}_1 = \mathcal{T}$ (where $\mathcal{T}(A) = \text{Tr}(A)|0\rangle\langle 0|$) then $\mathcal{C}$ becomes the usual erasure channel with erasure probability $1 - \lambda$, whose quantum capacity is zero for $\lambda \le 1/2$. However, an application of Theorem 1 shows that as $\mathcal{C}_1$ is changed from $\mathcal{T}$ to one of several different zero capacity qubit channels, coherent information of $\mathcal{C}$ becomes positive for all $\lambda > 0$.

Theorem 1 can be applied to the incomplete erasure channel $\mathcal{C}(\rho) = \lambda\rho \oplus (1-\lambda)\mathcal{C}_1(\rho)$[47] whose output is split into two orthogonal subspaces (see Fig. 4). The channel's input is sent unchanged to the first subspace with probability $\lambda$, else it is sent via a noisy channel $\mathcal{C}_1$ to the second subspace. This channel $\mathcal{C}$ is relevant for describing noise in experiments where the channel user knows if noise has acted or not.

Suppose the incomplete erasure channel $\mathcal{C}$ has a qubit input and $\mathcal{C}_1$ is any zero quantum capacity qubit channel with a qubit environment[63]. Any such qubit channel $\mathcal{C}_1$ has a noise parameter $0 \le p \le 1/2$; where at $p = 0$, $\mathcal{C}_1$ erases its input by taking it to a fixed pure state, making $\mathcal{C}$ a regular erasure channel. This regular erasure channel has zero coherent information, i.e., $\mathcal{Q}^{(1)}(\mathcal{C}) = 0$ when $\lambda$ is below a threshold $\lambda_0 = 1/2$[69]. As the noise in $\mathcal{C}_1$ is decreased by continuously increasing $p$, this threshold is expected to decrease continuously. While ordinary numerics may seem to confirm this expectation, simple use of Theorem 1 shows that for any $p > 0$, $\mathcal{Q}^{(1)}(\mathcal{C}) > 0$ for any $\lambda > 0$, i.e., an arbitrarily small increase in $p$ from zero shifts the threshold value from $\lambda_0 = 1/2$ to $\lambda_0 = 0$. This discontinuous shift doesn't appear in standard numerics because for small $p$, $\mathcal{Q}^{(1)}(\mathcal{C})$ can be as small as $O(e^{-1000})$[47], a number much beyond ordinary numerical precision. Such discontinuous shifts reveal an unexpected behavior: a channel $\mathcal{C}_1$ which can't send quantum information on its own, i.e., it has no quantum capacity, can nonetheless assist an incomplete erasure channel in sending quantum information. Such assistance was found previously for two specific qubit channels $\mathcal{C}_1$ using arguments tailored for those specific channels[44,47]. Our argument here generalizes those results and points to log-singularities as a generic mathematical cause behind this assistance. This assistance is particularly intriguing because it occurs for a wide variety of zero capacity qubit channels $\mathcal{C}_1$ but doesn't occur for arbitrary zero capacity channels. For instance, when $\mathcal{C}_1$ is a zero quantum capacity erasure channel with arbitrary input dimension and erasure probability $\mu \ge 1/2$, $\mathcal{Q}^{(1)}(\mathcal{C}) = 0$ for $0 \le \lambda \le 1 - 1/(2\mu)$.

To check if a channel $\mathcal{B}: a \mapsto b$ has strictly positive coherent information one may numerically find an input density operator $\rho$ for which the entropy difference $\Delta(\mathcal{B}, \rho) > 0$. This numerical search can be unreliable because $\Delta(\mathcal{B}, \rho)$ is generally non-convex in $\rho$ and $\Delta(\mathcal{B}, \rho)$ can be affected by log-singularities. The search can also be expensive for channels with large input and output dimensions. A corollary of Theorem 1 gives an algebraic result revealing a wide and simple set of channels with large output dimension and nonzero coherent information:

**Corollary 1**. Any channel $\mathcal{B}$ has $\mathcal{Q}^{(1)}(\mathcal{B}) > 0$ if its input dimension $d_a > 1$ and output dimension $d_b > d_a(d_c - 1)$.

This result is easy to apply: given a channel one simply uses its dimensions to check if the channel satisfies the conditions of Corollary 1. For instance, Corollary 1 implies that any channel whose output dimension is larger than its input dimension has a strictly positive $\mathcal{Q}^{(1)}$ whenever the channel's environment is a qubit, i.e., $d_c = 2$.

Qubit channels are extremely useful for characterizing noise in experiments. Despite the vast body of work dedicated to studying the capacity of qubit channels[63,70–74], a basic question has remained open: when does the complement of a qubit channel have nonzero quantum capacity? Corollary 1, in conjunction with prior work[63], answers this question. If the complement of a qubit channel has output dimension 1 or 2, then conditions under which this complement has nonzero capacity can be found in ref. [63]; for all remaining cases, Corollary 1 (see Supplementary Note 4) shows that the complement has strictly positive coherent information and quantum capacity. This positivity result contains as a special case the results of[65] which showed that any qubit Pauli channel $\mathcal{B}$ with $d_c = 3$ or 4 has a complement with positive channel coherent information.

**Nonadditivity of coherent information**. A mechanism based on log-singularities can give rise to nonadditivity of $\mathcal{Q}^{(1)}$. For two (possibly different) quantum channels $\mathcal{B}_1$ and $\mathcal{B}_2$, let

$$\mathcal{Q}^{(1)}(\mathcal{B}_1) = \Delta(\mathcal{B}_1, \rho_{a1}^*), \quad \text{and} \quad \mathcal{Q}^{(1)}(\mathcal{B}_2) = \Delta(\mathcal{B}_2, \rho_{a2}^*), \quad (5)$$

for some density operators $\rho_{a1}^*$ and $\rho_{a2}^*$ and let $\mathcal{B} := \mathcal{B}_1 \otimes \mathcal{B}_2$. Choose $\rho_a(\epsilon)$ at the input of $\mathcal{B}$ with the property that $\rho_a(0) = \rho_{a1}^* \otimes \rho_{a2}^*$ and $S_b(\epsilon)$ has a stronger $\epsilon$ log-singularity than $S_c(\epsilon)$, then a small enough increase in $\epsilon$ from zero will increase $\Delta(\epsilon)$ from $\mathcal{Q}^{(1)}(\mathcal{B}_1) + \mathcal{Q}^{(1)}(\mathcal{B}_2)$ by $|O(\epsilon \log \epsilon)|$ indicating a strict inequality in (1). This log-singularity-based mathematical mechanism responsible for nonadditivity requires $S_b(\epsilon)$ to have an $\epsilon$ log-singularity. As stated earlier, this requirement can be satisfied if $\rho_a(0)$ has less than full rank. A condition satisfied by several channels with zero and nonzero coherent information. This mechanism will now be used in an explicit instance of nonadditivity using two channels, one with zero and another with large positive coherent information.

To present this instance of nonadditivity, we introduce a low-noise qutrit channel $\mathcal{B}_1$ whose coherent information is comparable to that of a qutrit identity channel. This channel's superoperator $\mathcal{B}_1(\rho) = \text{Tr}_{c1}(J_1 \rho J_1^\dagger)$ comes from an isometry $J_1: a1 \mapsto b1 \otimes c1$ of the form,

$$J_1|0\rangle = \sqrt{s}|00\rangle + \sqrt{1-s}|11\rangle, \quad J_1|1\rangle = |21\rangle, \quad \text{and} \\ J_1|2\rangle = |20\rangle, \quad (6)$$

where $0 \le s \le 1$. Since an exchange of $s$ with $1 - s$ can be achieved by local unitaries in $a1$, $b1$, and $c1$, we restrict ourselves to $0 \le s \le 1/2$. The channel coherent information $\mathcal{Q}^{(1)}(\mathcal{B}_1)$ is given by its entropy difference $\Delta(\mathcal{B}_1, \rho_{a1}^*)$ where $\rho_{a1}^* = (1 - w)[0] + w[1]$, $0 < w < 1$ (see Supplementary Note 5). At $s = 0$, $\mathcal{Q}^{(1)}(\mathcal{B}_1) = 1$ and decreases monotonically with the noise parameter $s$ to become $\simeq 0.695$ at $s = 1/2$. These values of $\mathcal{Q}^{(1)}(\mathcal{B}_1)$ bound the quantum capacity of $\mathcal{B}_1$ from below and they are comparable to the quantum capacity, $\log_2 3$, of the qutrit identity channel.

A log-singularity-based argument stated earlier shows that using $\mathcal{B}_1$ in parallel with $\mathcal{B}_2$, a zero quantum capacity qubit amplitude damping channel with damping probability $p \ge 1/2$, results in nonadditivity, i.e., a strict inequality in (1) for all $0 < s \le 1/2$ and $1/2 \le p < \bar{p}(s)$ (see Fig. 5 and Methods Section). This nonadditivity has several interesting features. First, it shows

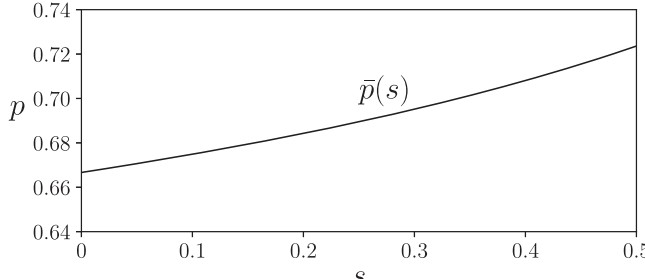

**Fig. 5 Nonadditivity in a low-noise channel.** For the low-noise qutrit channel $\mathcal{B}_1$ with noise parameter $0 < s \le 1/2$, $0.695 \le \mathcal{Q}^{(1)}(\mathcal{B}_1) < 1$ and qubit amplitude damping channel $\mathcal{B}_2$ with damping probability $1/2 \le p \le 1$, $\mathcal{Q}^{(1)}(\mathcal{B}_2) = \mathcal{Q}(\mathcal{B}_2) = 0$, we find $\mathcal{Q}^{(1)}(\mathcal{B}_1 \otimes \mathcal{B}_2)$ to be strictly larger than $\mathcal{Q}^{(1)}(\mathcal{B}_1) + \mathcal{Q}^{(1)}(\mathcal{B}_2)$ when $1/2 \le p \le \bar{p}(s)$ with $\bar{p}(s)$ plotted above.

the existence of previously unknown nonadditivity when using a low-noise channel. Second, the nonadditivity reported here fosters a more nuanced understanding of quantum capacity. Even in a setting using very simple low-dimensional channels, the quantum capacity provides an incomplete description of a channel's ability to send quantum information. As shown here, despite having no quantum capacity, the qubit amplitude damping channel $\mathcal{B}_2$ does posses a separate ability to assist transmission of quantum information when used in parallel with a simple qutrit channel $\mathcal{B}_1$. Third, the nonadditivity here is robust against amplitude damping noise: additional amplitude damping noise beyond $p = 1/2$ does not immediately destroy this nonadditive effect which survives till $p < \bar{p}(s)$. Fourth, this instance of nonadditivity has a very wide range: it is present over the entire parameter space of the qutrit channel $\mathcal{B}_1$, except at a single point $s = 0$. Fifth, numerical techniques, which are commonly used to find nonadditivity, can easily miss this wide range of nonadditivity which appears because of changes in entropy caused by log-singularities.

## Discussion

We have discussed logarithmic (log) singularities that occur quite generally in the von-Neumann entropy of any density operator being moved linearly from the boundary to the interior of the set of density operators. In the region where this singularity occurs, it dominates the behavior of the von-Neumann entropy. This kind of dominance can be used to extract insights about physics which depends on this entropy. We have investigated the physics of sending quantum information. Our investigation leads to an insight that log-singularities act as a mathematical source behind both positivity and nonadditivity in a channel's coherent information. An analysis of log-singularities could potentially be useful in other areas where the von-Neumann entropy plays a central role. One area of this type is the study of continuous variable channels. Capacities of these channels remain an active area of research[75–81], and these capacities also display a variety of exotic behaviour[82], including superactivation[83–85]. Extending our log-singularity ideas to investigate such exotic behavior would be an interesting direction of future work.

Checking if any general channel has nonzero or zero quantum capacity is a fundamental but hard problem. While some general methods have been proposed to solve this problem[61], they are not always easy to apply and don't necessarily lead to new channels with zero or positive quantum capacity. By contrast, the algebraic log-singularity-based method proposed here is easy to apply and it unearths a variety of channels with positive quantum capacity. Using it, we give Theorem 1 which reveals certain general conditions for strict positivity of a channel's coherent information.

Corollary 1 of Theorem 1 unearths a wide variety of channels with strictly positive coherent information. Corollary 1 only makes use of a channel's dimensions. Extending this corollary, for example by showing $\mathcal{Q}^{(1)}(\mathcal{B}) > 0$ for some $d_b > d_c$, would be an interesting direction of future research. Yet another direction would be to supplement our mathematical log-singularity reasoning with more physical arguments. Such reasoning may help explain the positivity of $\mathcal{Q}^{(1)}$ found in the incomplete erasure channel and clarify why the simplest zero capacity channels behave differently from others when used as part of the incomplete erasure channel. This clarification may provide further insights into the transmission of quantum information. Another source of insight may be a quantitative analysis of bounds on the quantum capacity[59,64,86–91] of the incomplete erasure channel or some other channel where a log-singularity-based mechanism is responsible for strict positivity of the channel's coherent information. In certain cases, log-singularities can dominate continuity-based bounds on the coherent information and it would be interesting to see if such effects also appear in continuity bounds on a channel's quantum capacity[86].

Using log-singularities, we have shown that the coherent information and quantum capacity of several channels is nonzero. It follows that the two-way quantum capacity[30] and the private capacity[36] of these same channels are also nonzero. These observations comes from the simple fact that the quantum capacity of a channel is a lower bound on the channel's private and two-way quantum capacities. These other capacities are even less understood than the quantum capacity and our log-singularity-based analysis could prove useful in their investigation. For instance, the reverse coherent information[92], which is a lower bound on the two-way quantum capacity, may yield to a log-singularity-based analysis, similar to the one performed here. Admittedly, our log-singularity-based method for showing positivity of capacity does not solve the general problem of finding all channels with strictly positive capacity. Results concerning unbounded nonadditivity[49,50] temper hopes about the existence of an easy to apply but a completely general method for checking positivity of the quantum capacity. Our work nonetheless points out that such tempering need not hinder progress in finding new, interesting, and potentially insightful instances of channels with strictly positive quantum capacity.

Another aspect of our findings is how changes in the von-Neumann entropy caused by log singularities is a mathematical mechanism responsible for nonadditivity. Prior search for such mechanisms have focussed on the structure of special channels[42,56,57] used for obtaining nonadditivity or on the use of certain tailored quantum codes[45,60,93]. We open another direction by showing how a fundamental property of the von-Neumann entropy can lead to nonadditivity. This log-singularity property can be analyzed to find nonadditivity in channels with large, small, or no quantum capacity. The algebraic nature of this analysis allows us to identify nonadditivity over wide ranges of a channel's parameter, without the need for traditional numerics[43,46,67]. While our work opens one path, it is not the only path forward. We leave open the exciting but challenging possibility of finding other mathematical and physical principles that may explain non-additive effects in quantum information science.

Unlike prior work, nonadditivity of the coherent information reported here occurs in a low-noise channel. From a fundamental physics perspective, nonadditivity using the product of one low-noise channel with another low-dimensional but zero capacity qubit amplitude damping channel is surprising because it implies that even in such a simple setting the quantum capacity of the amplitude damping channel does not fully characterize its resourcefulness for sending quantum information.

This simple example adds to the collection of exotic channels from which further physics can be extracted. In principle, this low-dimensional and low-noise channel can be experimentally realized. Given the practical relevance of low-noise channels, our finding of nonadditivity in such channels suggests that non-additivity is not just a fundamental curiosity but a potential resource for quantum technologies.

## Methods

**Proof of Theorem 1.** A log-singularity-based mechanism responsible for making a channel's coherent information strictly positive is the key ingredient in the proof of Theorem 1. Assume $d_c < d_b$ and $\mathcal{B}$ maps some pure state $[\psi]_a$ to an output $\mathcal{B}([\psi]_a)$ of rank $d_c$. Consider an input density operator $\rho_a(\epsilon)$ of the form in eq. 2 where $\hat{\rho}_a$ is the pure state $[\psi]_a$ and $\sigma_a = I_a/d_a$. The outputs $\rho_b(\epsilon)$ and $\rho_c(\epsilon)$ have the form in (3) where $\hat{\rho}_b$ and $\hat{\rho}_c$ have the same rank $d_c$, $\sigma_c$, and $\sigma_b$ have ranks $d_c$ and $d_b$, respectively (see para 3 in Methods Section). As a result $S_b(\epsilon)$ has an $\epsilon$ log-singularity while $S_c(\epsilon)$ doesn't, consequently $\mathcal{Q}^{(1)}(\mathcal{B}) > 0$. The absence of an $\epsilon$ log-singularity in $S_c(\epsilon)$ follows from the fact that at $\epsilon = 0$, $\rho_c(\epsilon)$ is a rank $d_c$ operator $\hat{\rho}_c$. To notice the presence of an $\epsilon$ log-singularity in $S_b(\epsilon)$, it is helpful to rewrite $\rho_b(\epsilon)$ in eq. 3, as

$$\rho_b(\epsilon) = \hat{\rho}_b + \epsilon \omega_b, \tag{7}$$

where

$$\omega_b := \sigma_b - \hat{\rho}_b. \tag{8}$$

At $\epsilon = 0$, $\rho_b(\epsilon)$ is $\hat{\rho}_b$, which has $d_b - d_c$ zero eigenvalues. Corresponding to these zero eigenvalues is an eigenspace of dimension $d_b - d_c > 0$. Let $P_0$ be a projector onto this eigenspace and

$$\tilde{\omega}_b := P_0 \omega_b P_0 = P_0 \sigma_b P_0. \tag{9}$$

Since $\sigma_b$ has rank $d_b$, the operator $\tilde{\omega}_b$ is positive definite on the support of $P_0$, thus $\tilde{\omega}_b$ has $(d_b - d_c)$ strictly positive eigenvalues $\{e_i\}$. Elementary results from perturbation theory (for instance see Section 5.2 in ref. [94]) show that all $d_b - d_c$ zero eigenvalues of $\rho_b(\epsilon)$ at $\epsilon = 0$ become nonzero for positive $\epsilon$, and to leading order in $\epsilon$ these eigenvalues increase linearly such that the $i^{th}$ such eigenvalue is simply $\epsilon e_i$. As a result, $S_b(\epsilon)$ has an $\epsilon$ log-singularity.

**Nonadditivity.** A log-singularity-based mechanism is responsible for nonadditivity (1) when $\mathcal{B}_1$ and its complement $\mathcal{C}_1$ are channels defined by $J_1$ in (6) and $\mathcal{B}_2$, along with its complement $\mathcal{C}_2$, are defined an isometry $J_2 : \mathcal{H}_{a2} \mapsto \mathcal{H}_{b2} \otimes \mathcal{H}_{c2}$ of the form,

$$J_2|0\rangle = |00\rangle, \quad J_2|1\rangle = \sqrt{1-p}|10\rangle + \sqrt{p}|01\rangle. \tag{10}$$

Here $\mathcal{B}_2$ represents a qubit amplitude damping channel with damping probability $p$. We shall be interested in the parameter region $1/2 \le p \le 1$ where $\mathcal{B}_2$ is anti-degradable and $\mathcal{Q}^{(1)}(\mathcal{B}_2) = \mathcal{Q}(\mathcal{B}_2) = 0$[63]. Consider the channel pair $\mathcal{B} = \mathcal{B}_1 \otimes \mathcal{B}_2$, $\mathcal{C} = \mathcal{C}_1 \otimes \mathcal{C}_2$, with channel input

$$\rho_a(\epsilon) = (1-w)[00]_a + w[\chi_\epsilon]_a, \tag{11}$$

where

$$|\chi_\epsilon\rangle_a = \sqrt{1-\epsilon}|10\rangle_a + \sqrt{\epsilon}|21\rangle_a, \tag{12}$$

$0 \le \epsilon \le 1$, and $w$ is chosen such that at $\epsilon = 0$,

$$\rho_a(0) = ((1-w)[0]_{a1} + w[1]_{a1}) \otimes [0]_{a2} = \rho_{a1}^* \otimes \rho_{a2}^*, \tag{13}$$

i.e., $\Delta(0) = \mathcal{Q}^{(1)}(\mathcal{B}_1) + \mathcal{Q}^{(1)}(\mathcal{B}_2)$. For any $\epsilon > 0$ and $s > 0$, an eigenvalue $((1-p)w)\epsilon$ of $\rho_b(\epsilon)$ and an eigenvalue $(pkw)\epsilon$,

$$k = (1-s)(1-w)/(w + (1-s)(1-w)) < 1, \tag{14}$$

of $\rho_c(\epsilon)$ increases linearly from zero to leading order in $\epsilon$. Thus $S_b(\epsilon)$ has an $\epsilon$ log-singularity of rate $(1-p)w$ and $S_c(\epsilon)$ has a $\epsilon$ log-singularity of rate $pkw$. The $\epsilon$ log-singularity in $S_b(\epsilon)$ is stronger when $p < \bar{p}(s) = 1/(1+k)$ (plotted in Fig. 5). This stronger singularity implies nonadditivity, i.e., a strict inequality in eq. 1.

## Data availability
No data sets were generated during this study.

## Code availability
Source code for the plots in this study are available on a Github repository, https://github.com/vsiddhu/logSing.

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

## Acknowledgements
The author thanks Robert B. Griffiths, Graeme Smith, Costin Bădescu, Yang Gao, Michael Widom, and Mark M. Wilde for their useful comments. This work was partially supported by NSF CAREER Award CCF 1652560 and NSF Grant PHY 1915407.

## Author contributions
The author performed research and wrote this paper.

## Competing interests
The author declares no competing interests.
