## [Peer Review File · Nature Communications]

Reviewers' Comments:

Reviewer #1:

Remarks to the Author:

In this paper, the authors introduce log-singularities, an analytical technique. This technique is successfully applied to find that the coherent information of very simple channels is non-additive and to finalize the classification of qubit channels with respect to whether or not their complements have zero capacity. The results are solid and increase our little understanding of channel capacities. While the advances are rather technical, given the little progress encountered in this arena in the past years, I would suggest acceptance. My comments mostly regard the presentation of the material:

- Not all relevant material seems to be referenced. For instance:
 - * the recent work of Filippov "Capacity of trace decreasing quantum operations and superadditivity of coherent information for a generalized erasure channel" on JPA is missing.
 - * "Capacity Bounds via Operator Space Method", "Capacity Estimates via Comparison with TRO Channel" from the group of Marius Junge and related work.
- The authors do a good job placing in context their findings. However, some of the arguments seem to hit a straw man. For instance, it is unclear who had a long-held belief that non-additive effects appear only when two noisy channels are used in parallel (page 4). One can consider N_1 a ppt pbite channel tensored with a large identity and N_2 an erasure channel tensored also with a large identity. Both N_1 and N_2 clearly have a large capacity and still, we can observe non-additivity. Of course, this is a trivial example, while the example in the paper is not and is also low dimensional. But still, I would rephrase the language of long-held beliefs by a discussion of open problems. Unless these beliefs were indeed conjectured somewhere, but then I would ask the authors to add the corresponding reference.
- I find confusing the notation in section 2 in the main text. ρ depends on ϵ , but the function of the entropy has ρ without reference to ϵ .
- I find vague calling log-singularities a mechanism, following the definition they are a property of a one-parameter family of density operators. It is not defined what is the "log-singularities mechanism".
- The term singularity itself is a little bit misleading. The coherent information is continuous with respect to trace distance and also its maximum with respect to diamond distance. This means that the effects of these singularities both in terms of states as well as for channels are bounded by Alicki-Fannes-Winter inequality and can not be too wild, I am thinking of the example that illustrates the main text with the jump to positive capacity when $\lambda > 0$. Maybe the authors want to keep the name singularity, but should comment on how it relates to continuity.
- At the bottom of page 3, I understand that B_2 should be B_1 .
- The Supplementary information needs further revision. I find it confusing having appendices for the supplementary which is not very long. The structure of the SI sections is strange, with the many sections overly linked. Also, SI-2 is a paragraph without ending; the last sentence reads "Concisely denote the entropy bias of B at ρ by $\Delta(\epsilon)$ ". I would suggest to either add a couple of paragraphs before SI-1 explaining how to navigate the supplementary information or to add it to the end of SI-1.

Reviewer #2:

Remarks to the Author:

The authors of the manuscript titled "Entropic singularities give rise to quantum transmission" showed how the log singularity of a channel can lead to the non-additivity of coherent information. While it is commonly believed that non-additivity occurs with two very noisy channels, the authors showed that it can occur with a channel with a large quantum capacity and another channel with vanishing quantum capacity. Unlike many previous papers where non-additivity is numerically demonstrated without much insight, this paper provides an interesting insight into how the non-additivity can arise by using log singularity of quantum entropy. However, I think the manuscript needs some improvements in terms of presentation and clarity. While I believe this manuscript may merit publication in Nature Communications upon suitable revisions, I do not think it does in its current form.

Comments:

--I think the example in the caption of Fig. 1 is worth being discussed in the main text. While Fig. 1 is referred to in the main text, I didn't expect to see an example which is not discussed in the main text. Although it is a simple example which can be introduced in a self-contained way in a figure caption, I still think it might be better to feature this example in the main text as well so the readers can expect what to take away from Fig. 1.

--On the bottom of page 3 after $s=1/2$, should $\mathcal{Q}^{\{1\}}(\mathcal{B}_{\{2\}})$ and $\mathcal{B}_{\{2\}}$ be $\mathcal{Q}^{\{1\}}(\mathcal{B}_{\{1^*\}})$ and $\mathcal{B}_{\{1^*\}}$ instead?

--The caption of Fig. 3 is not self-contained since $\mathcal{B}_{\{1\}}$ and $\mathcal{B}_{\{2\}}$ are not defined in the caption. Also, I think $\mathcal{Q}^{\{1\}}(\mathcal{B}_{\{1\}}) = \mathcal{Q}(\mathcal{B}_{\{1\}})$ is a typo and should be $\mathcal{Q}^{\{1\}}(\mathcal{B}_{\{2^*\}}) = \mathcal{Q}(\mathcal{B}_{\{2^*\}})$

--Also, while I understand the proof of the proposition in the caption of Fig. 3 is given in the supplementary material, I think it could help bringing some key features in the proof to the main text, such as the rates of log singularity of relevant entropies.

--Fig. 4 is based on the example in Fig. 1. However, it was not clear to me if the example states in Fig. 1 actually stem from some quantum channel \mathcal{B} . In other words, it was not clear what the channel \mathcal{B} is in the caption of Fig. 4.

--On the bottom of page 4, I think " $\mathcal{Q}(\mathcal{B}) >$ is new" has to be " $\mathcal{Q}(\mathcal{B}) > 0$ is new".

Reviewer #3:

Remarks to the Author:

The authors analyse from an optimisation-theoretic angle the coherent information function on specific low-dimensional quantum channels. Based on a concept termed "epsilon-log-singularity" coming from non-full rank optimisers, this leads to a novel sufficient criteria for when the quantum capacity of quantum channels is non-zero. The condition is quite neat because it only depends on the dimension of the quantum channels analysed.

Similarly, the coherent information is shown to be non-additive on the product of two particular low noise channels (which does not directly make a statement about quantum capacities?).

The approach presented is conceptually a bit different from previous work that either performed some type of clever brute force numerics for the non-convex optimisation of the coherent information, or alternatively directly constructed quantum codes in order to quantify the information-theoretic power of quantum channels. The methods introduced come with some advantages outlined in the submission and as such the derived results are definitely interesting for the quantum communication theory community. However, overall I am not convinced that Nature Communications is the right venue for this work to be published. The reasons are as follows:

-The results are technical in nature and it is, e.g., not very transparent to distill what are the new bounds derived on the quantum capacity of quantum channels. As such, it is not clear at this point what are the far-reaching contributions for quantum information theory that are of broader interest.

-The abstract is super vague and to me it was not clear what the actual main results are before diving into the main body of the paper. At the same time, the abstract is overselling about the results derived. For example, what "long-standing open problem" is solved? (Going to pages 2/6, I assume this is the question of "when does the complement of a qubit channel have non-zero quantum capacity"?)

-I am not sure what "for two decades, non-additivity was believed to occur in very noisy devices" exactly means. It is true that prominent examples of non-additivity occur for noisy devices, but apart from that I am not sure I follow the argument?

-I had to read two full pages about the landscape of quantum communication theory before I learnt anything about the results derived here. At the same time the explanations and pictures given in the main text are very simple and to actually understand any of the inner workings of the proofs one has to refer to the supplementary material (that is then again broken up in a main text plus appendices). This makes it cumbersome to understand the results; as presented they simply do not seem to fit the journal template.

To conclude, the results are interesting but are presented somewhat chaotically and are not of sufficiently broad interest. Nevertheless, I judge that the submission would make a good theory paper and I encourage the authors to submit a rearranged manuscript to a more mathematical / specialised journal. I expect the results to generate some good follow-up work in the area of quantum communication theory.

Additional comments:

-What exactly does the non-additivity of the coherent information on the product of two different quantum channels tell us about the quantum capacity and its additivity properties? Note that the non-additivity of the coherent information on products of the same channels directly gives bounds on the quantum capacity of said channels. Can similar statements be concluded from the author's result?

Response addressing points raised by reviewer one.

1.

Rev. Not all relevant material seems to be referenced. For instance:

- * the recent work of Filippov "Capacity of trace decreasing quantum operations and superadditivity of coherent information for a generalized erasure channel" on JPA is missing.
- * "Capacity Bounds via Operator Space Method", "Capacity Estimates via Comparison with TRO Channel" from the group of Marius Junge and related work.

Ath. In addition to all suggested papers above, various other papers,

- ``Private and quantum capacities of more capable and less noisy quantum channels`` by Watanabe,
- ``Continuity of Quantum Channel Capacities`` by Leung and Smith
- ``Channel covariance, twirling, contraction and some upper bounds on the quantum capacity`` by Ouyang
- ``Uniform Additivity in Classical and Quantum Information`` by Cross, Li, and Smith ,
- ``Useful States and Entanglement Distillation`` by Leditzky, Datta, and Smith,
- ``Quantum Flags and New Bounds on the Quantum Capacity of the Depolarizing Channel`` by Fanizza, Kianvash, and Giovannetti,

have also been cited in relevant parts of the paper.

2.

Rev. The authors do a good job placing in context their findings. However, some of the arguments seem to hit a straw man. For instance, it is unclear who had a long-held belief that non-additive effects appear only when two noisy channels are used in parallel (page 4). One can consider N_1 a ppt pbit channel tensored with a large identity and N_2 an erasure channel tensored also with a large identity. Both N_1 and N_2 clearly have a large capacity and still, we can observe non-additivity. Of course, this is a trivial example, while the example in the paper is not and is also low dimensional. But still, I would rephrase the language of long-held beliefs by a discussion of open problems. Unless these beliefs were indeed conjectured somewhere, but then I would ask the authors to add the corresponding reference.

Ath. To address the first and third part of this comment, wording about prior beliefs have been removed, not only from the Introduction but also from all other parts of the manuscript. As suggested, discussion in the Introduction about prior beliefs has been replaced with a discussion of prior work in light of which non-additivity in low-noise channels is surprising. A variety of open problems arising from this work are stated in the Discussion section.

The second part of the above comment distinguishes non-additivity in low-noise channels found in this work with other examples modelled on previous PPT and erasure channel cases. To clarify the kind of difference being drawn above we explicitly define what we mean by a low-noise channel (one whose coherent information is comparable to an identity channel of the same input dimension) in the abstract,

introduction, and results sections.

3.

Rev. I find confusing the notation in section 2 in the main text. ρ depends on ϵ , but the function of the entropy has ρ without reference to ϵ .

Ath. This notation has been revised by replacing ρ with ρ of ϵ

4.

Rev. I find vague calling log-singularities a mechanism, following the definition they are a property of a one-parameter family of density operators. It is not defined what is the "log-singularities mechanism".

Ath. We agree, a log-singularity is not a mechanism but a property. Changes in the von-Neumann entropy caused by log-singularities are a mathematical mechanism responsible for both positivity and non-additivity of the coherent information. To avoid confusion, we have suitably modified all parts of the text as follows:

- In the introduction, where this mechanism is first referenced, we clearly state the mechanism's relation to log-singularities.
- In the Results section we introduce topical sub-headings for positivity and non-additivity. In each heading we first give a detailed presentation of the mechanism and its relationship to log-singularities.
- The mechanism, having been clearly defined in the results section, is referenced in elsewhere as a log-singularity **based** mechanism.

5.

Rev. The term singularity itself is a little bit misleading. The coherent information is continuous with respect to trace distance and also its maximum with respect to diamond distance. This means that the effects of these singularities both in terms of states as well as for channels are bounded by Alicki-Fannes-Winter inequality and can not be too wild, I am thinking of the example that illustrates the main text with the jump to positive capacity when $\lambda > 0$. Maybe the authors want to keep the name singularity, but should comment on how it relates to continuity.

Ath. We retain the term log-singularity and motivate this usage right after defining the term. The motivation comes from the von-Neumann entropy's derivative behaving logarithmically and tending to infinity under certain conditions involving a parameter being made small.

To address the second part of this comment, right after motivating our usage of the term log-singularity, we recognize the entropy itself is continuous and state the relationship between continuity bounds and log-singularities. This relationship, explored fully in a new Supplementary Note using not only the Alicki-Fannes-Winter inequality but also the Fannes-Audenaert inequality, is essentially that changes in the continuity upper bound can also be logarithmic in cases where a log-singularity is present.

Along these continuity argument lines, additional bounds on quantum capacities in terms of diamond norms can be constructed [1]. They are not always optimal [2], but in the case of incomplete erasure channels, they

may very well give more insight than other bounds. Doing proper justice to such bounds in a separate manuscript is a potentially fruitful research direction motivated in the Discussion section.

[1] - ``Continuity of Quantum Channel Capacities``. Leung and Smith. Comm. Math. Phys. 292, 201-215 (2009)

[2] - ''Quantum and private capacities of low-noise channels'', Leditzky, Leung, and Smith Phys. Rev. Lett. 120, 160503 (2018)

6.

Rev. At the bottom of page 3, I understand that B₂ should be B₁.

Ath. Yes, this typographical error has been corrected.

7.

Rev. The Supplementary information needs further revision. I find it confusing having appendices for the supplementary which is not very long. The structure of the SI sections is strange, with the many sections overly linked. Also, SI-2 is a paragraph without ending; the last sentence reads "Concisely denote the entropy bias of B at rho by Delta(epsilon)". I would suggest to either add a couple of paragraphs before SI-1 explaining how to navigate the supplementary information or to add it to the end of SI-1.

Ath. We have revamped the supplementary information by moving some material to the main text, removing prior appendices, and reorganizing all the material into separate supplementary notes with descriptive headings. As a result of these changes the new supplementary notes contain closely related results, minimally linked with content in other supplementary notes. The second part of the comment was naturally addressed by the reorganization as the sentence in question was moved to the beginning of a paragraph in the main text. To address the third part of the comment, at the end of Supplementary Note 1, Preliminaries, we add a short description of various supplementary notes and their relationship to the main text or each other. We also give a brief description of the supplementary note at the beginning of each note.

response addressing points raised by reviewer two.

1.

Rev. I think the example in the caption of Fig. 1 is worth being discussed in the main text. While Fig. 1 is referred to in the main text, I didn't expect to see an example which is not discussed in the main text. Although it is a simple example which can be introduced in a self-contained way in a figure caption, I still think it might be better to feature this example in the main text as well so the readers can expect what to take away from Fig. 1.

Ath. We agree, at two places in the Results section of the main text, right after defining log-singularities and below eq. (4), we add a discussion of the example in Fig.1.

2.

Rev. On the bottom of page 3 after $s=1/2$, should $\mathcal{Q}^{\{1\}}(\mathcal{B}_{\{2\}})$ and $\mathcal{B}_{\{2\}}$ be $\mathcal{Q}^{\{1\}}(\mathcal{B}_{\{1^*\}})$ and $\mathcal{B}_{\{1^*\}}$ instead?

Ath. Yes, this typo has been fixed.

3.

Rev. The caption of Fig. 3 is not self-contained since $\mathcal{B}_{\{1\}}$ and $\mathcal{B}_{\{2\}}$ are not defined in the caption. Also, I think $\mathcal{Q}^{\{1\}}(\mathcal{B}_{\{1\}}) = \mathcal{Q}(\mathcal{B}_{\{1\}})$ is a typo and should be $\mathcal{Q}^{\{1\}}(\mathcal{B}_{\{2^*\}}) = \mathcal{Q}(\mathcal{B}_{\{2^*\}})$

Ath. We modify the caption of Fig. 3 (Fig. 5 in the current version) to make it more self-contained by briefly defining channels B_1 and B_2 . To address the second part of this comment, we correct the typo.

4.

Rev. Also, while I understand the proof of the proposition in the caption of Fig. 3 is given in the supplementary material, I think it could help bringing some key features in the proof to the main text, such as the rates of log singularity of relevant entropies.

Ath. As suggested, key features of the proof have been brought to the main text in two ways. First, we explicitly discuss the log-singularity based mechanism in the Results section. Second, we add a Methods section, where under a topical sub-heading, Non-additivity, we give full details supporting Fig 5 (previously Fig 3) including rates for log-singularities of relevant entropies.

5.

Rev. Fig. 4 is based on the example in Fig. 1. However, it was not clear to me if the example states in Fig. 1 actually stem from some quantum channel \mathcal{B} . In other words, it was not clear what the channel \mathcal{B} is in the caption of Fig. 4.

Ath. In the previous version, the example in Fig. 1 served the purpose of introducing log-singularities but was also designed in a way to aid the exposition of the log-singularity based mechanism responsible for positivity of a channel's coherent information. The states in Fig. 1 do come from a complementary pair of quantum channels. In this version, the channel isometry and channel inputs which give rise to these states have been added (see discussion containing eq. (4)). The caption in Fig. 3 (previously Fig. 4) has been modified to indicate the channel under discussion. The rate of log-singularities in both Figures have been rescaled by $12/13$ in an effort to streamline notation with new equations (2) and (3), moved here from the previous Supplementary Notes to smoothen this new exposition.

6.

Rev. On the bottom of page 4, I think " $\mathcal{Q}(\mathcal{B}) > 0$ is new" has to be " $\mathcal{Q}(\mathcal{B}) > 0$ is new".

Ath. This typo has been corrected

response addressing points raised by reviewer three

1.

Rev. The results are technical in nature and it is, e.g., not very transparent to distill what are the new bounds derived on the quantum capacity of quantum channels. As such, it is not clear at this point what are the far-reaching contributions for quantum information theory that are of broader interest.

Ath. As essentially pointed out by the first and second reviewer, we not only make various technical but also novel and insightful contributions on important problems about which the research community knows very little, despite two decades of work. In agreement with these reviewer's sentiments, we believe the quality and quantity of these contributions are sufficient to justify publication, however, our results are not purely technical in nature and do make a variety of far reaching statements about fundamental quantities, including a newly added statement about the quantum capacity discussed next.

This new statement about the quantum capacity, added to the Introduction, Results and Discussion sections, stems from our example of non-additivity which shows that even in a setting using the product of a simple low-noise (and even low-dimensional) channel and the well known qubit amplitude damping channel, the quantum capacity does not provide a complete description of the qubit amplitude damping channel's resourcefulness to send quantum information. This finding fosters a more nuanced understanding of quantum capacities, beyond a common idea that 'a capacity quantifies a (quantum) channel's potential for communication' [3].

A noisy quantum channel not only models communication but also storage and processing of information, all these key fields constitute quantum information theory. As pointed out in the main text of our current and previous draft, our central contributions to all these fields are new log-singularity based mechanisms responsible for positivity and non-additivity of a quantum channel's coherent information.

An even broader point of appeal stated in the main text is the fundamental nature of the von-Neumann entropy and how log-singularities, introduced here, can dominate this entropy and the physics which may depend on it. Analysis of such singularities are expected to be useful in other physics problems as they have been shown here to be very useful to two foundational ones in quantum information: when and how much quantum information can be sent via noisy channels.

P.S. Part of this reviewer comment #1 about lower bounds overlaps with a longer comment, #6, about quantum capacities, and is thus addressed there in even more detail.

[3] - ``Quantum Channel Capacities``, G Smith,
2010 IEEE Information Theory Workshop, DOI: 10.1109/CIG.2010.5592851

2.

Rev. The abstract is super vague and to me it was not clear what the actual main results are before diving into the main body of the paper. At the same time, the abstract is overselling about the results derived. For example, what ``long-standing open problem`` is solved? (Going to pages 2/6, I assume this is the question of ``when does the complement of a qubit channel have non-zero quantum capacity``?)

Ath. The journal guidelines [4] state ``The abstract--- which should be no more than 150 words long and contain no references--- should serve both as a general introduction to the topic and as a brief, non-technical summary of the main results and their implications.`` Nonetheless it has been modified in several ways to improve its appeal for the more technically inclined readers. For such readers,

- the technical term word `quantum channel` has been added next to the word quantum device;
- a discussion about belief has been replaced with previously unknown low-noise non-additivity result reported here;
- the term low-noise has been contrasted with high-noise, defined here in terms of the channel coherent information;
- the precise long-standing problem being solved has also been stated specifically address the comment above.

In the abstract, it is not possible to state all results from this work in detail and still fulfill the purpose of a having a short non-technical abstract. However, we tip our hat to various results about log-singularity based mechanisms, Theorems and corollaries about channels with non-zero quantum capacity, and results on incomplete erasure channels. Obviously details about these results are in various sections of this work.

[4] - <https://www.nature.com/ncomms/submit/article>

3.

Rev. I am not sure what ``for two decades, non-additivity was believed to occur in very noisy devices`` exactly means. It is true that prominent examples of non-additivity occur for noisy devices, but apart from that I am not sure I follow the argument?

Ath. This comment refers to the abstract. To address it, the prior wording in question has been changed to ``For two decades, non-additivity was known to occur in very noisy channels with channel coherent information much smaller than that of a perfect channel``. This wording clarifies what we mean by a very noisy channel and states what was known previously. Of course, more can be said about prior non-additivity results, and why the one here is surprising. For curious readers seeking more, we have a longer discussion in the main text.

4.

Rev. I had to read two full pages about the landscape of quantum communication theory before I learnt anything about the results derived here. At the same time the explanations and pictures given in the main text are very simple and to actually understand any of the inner workings of the proofs one has to refer to the supplementary material (that is then again broken up in a main text plus appendices). This makes it cumbersome to understand the results; as presented they simply do not seem to fit the journal template.

Ath. This work is aimed for both experts in quantum information and the broad Nature Communications readership. To achieve this aim we find it useful to follow Nature Communications guidelines [4] and add background material which may be familiar to those working on non-additivity but perhaps not to others in quantum information, and most likely not familiar to those outside the field. Even for those familiar with non-additivity, the background provides context for our results, which the first reviewer found useful, and thus we don't abridge the Introduction.

Reorganization of, and additions to this paper address the second part of this comment. Given the broad readership of this journal, we retain simple explanations and pictures for readers who may be uninitiated in this line of work. For those already initiated and seeking more details, we add new, and bring forward prior technical material from the previous Supplementary Information (SI) into the Results section and create a new Methods section as well.

Due to these changes the new SI does not have any appendices. The new Results section first introduces log-singularities in full generality, then gives two simple illustrations (with accompanying figures), and makes more advanced comments. Next, we begin with an illustration of the log-singularity based mechanism behind positivity. This illustration is followed by a concrete example which has two accompanying plots, and finally we introduce the general log-singularity based mechanism behind positivity and results which come from using this mechanism. We state the theorem in this work, proceed to its applications but return to prove the theorem in the Methods section where we give all crucial inner workings for that proof. A similar template is followed for non-additivity.

The background and simplicity in the previous version was designed to fit the journal's template. We have retained that as much as possible while providing technical details in a reader friendly way.

5.

Rev. To conclude, the results are interesting but are presented somewhat chaotically and are not of sufficiently broad interest. Nevertheless, I judge that the submission would make a good theory paper and I encourage the authors to submit a rearranged manuscript to a more mathematical / specialised journal. I expect the results to generate some good follow-up work in the area of quantum communication theory.

Ath. First two parts of this comment, aimed at this work's presentation and broad interest, were raised in two prior comments. A detailed response to these accompanies those prior comments. Here we simply agree with the last statement in this comment.

6. [Additional Comment]

Rev. What exactly does the non-additivity of the coherent information on the product of two different quantum channels tell us about the quantum capacity and its additivity properties? Note that the non-additivity of the coherent information on products of the same channels directly gives bounds on the quantum capacity of said channels. Can similar statements be concluded from the author's result?

Ath. We specially thank the reviewer for this additional comment. In general, a non-additivity of the coherent information can not only provide a quantitative lower bound on a channel's quantum capacity, but it can also give deep qualitative insights into the concept of the quantum capacity itself.

The quantum capacity is sometimes stated to capture a channel's potential to send quantum information (for instance see [3]). Incompleteness of such statements can be shown by non-additivity of two different channels with fully or partially characterized quantum capacities. The novel non-additivity found here makes precisely one such previously unknown statement: quantum capacity of the well known qubit amplitude damping channel does not completely specify its potential for quantum communication. Despite having no quantum capacity, this qubit channel is shown to possess a potential to aid transmission of quantum information when used along with a simple low-noise channel. In addition to this statement about quantum capacities, it is noteworthy that such a statement is being made here using very simple channels, a qubit channel and a low-noise and low-dimensional qutrit channel.